# Differential Impairment Mechanism of Sperm Production via Induction of miR-34c-Activated Apoptosis and Spermatogenesis Pathway in Diet-Induced Obesity and Resistant Mice and GC-1 Spg Cells

**DOI:** 10.3390/ijms25137451

**Published:** 2024-07-07

**Authors:** Mujiao Li, Qing Zhao, Siyu Wang, Yangyang Song, Lingling Zhai, Jian Zhao

**Affiliations:** 1Department of Pharmacology, Shenyang Pharmaceutical University, No. 103, Wenhua Rd., Shenhe District, Shenyang 110016, China; limumu1907@outlook.com (M.L.); zhaoqing017@outlook.com (Q.Z.); wsyra0118@outlook.com (S.W.); songyangyang0110@outlook.com (Y.S.); 2Department of Maternal, Child and Adolescent Health, School of Public Health, China Medical University, Shenyang 110122, China; llzhai@cmu.edu.cn

**Keywords:** male reproduction, high-diet-induced obesity, high-diet-induced obesity resistant, apoptosis, spermatogenesis

## Abstract

Male reproductive dysfunction is a clinical disease, with a large number of cases being idiopathic. Reproductive disorders have been found in obese (diet-induced obesity and diet-induced obesity-resistant) mice, but the mechanism behind the male reproductive dysfunction between them may be different. The purpose of this study was to explore the possible role and mechanism of *miR-34c* on sperm production in high-fat-diet-induced obesity-resistant (DIO-R) mice and GC-1 spg cells, which may differ from those in high-fat-diet-induced obesity (DIO) mice. In vivo and in vitro experiments were performed. *C57BL/6J* mice were fed a high-fat diet for 10 weeks to establish the DIO and DIO-R mouse model. GC-1 spg cells were used to verify the mechanism of *miR-34c* on sperm production. During in vivo experiments, sperm production damage was found in both DIO and DIO-R male mice. Compared to the control mice, significantly decreased levels of testosterone, LH, activities of acrosome enzyme (ACE), HAse, and activating transcription factor 1 (ATF1) were found in both DIO and DIO-R male mice (*p* < 0.05). Compared with the control group, the ratio of B-cell lymphoma-2 (Bcl-2)/bcl-2-associated X protein (Bax) in the DIO group was significantly decreased, and the expression level of cleaved caspase-3 was significantly increased (*p* < 0.05). Compared with the control group, the Bcl-2 protein expression level in the testes of the DIO-R group significantly decreased (*p* < 0.05). However, the Bax expression level increased. Thus, the Bcl-2/Bax ratio significantly decreased (*p* < 0.01); however, the factor-related apoptosis (Fas), Fas ligand (FasLG), cleaved caspase-8, caspase-8, cleaved caspase-3, and caspase-3 protein expression levels significantly increased (*p* < 0.05). Compared with the DIO group, in DIO-R mice, the activities of ACE, ATF1, Bcl-2, and Bcl-2/Bax’s spermatogenesis protein expression decreased, while the apoptosis-promoting protein expression significantly increased (*p* < 0.05). During the in vitro experiment, the late and early apoptotic ratio in the *miR-34c* over-expression group increased. *MiR-34c* over-expression enhanced the expression of apoptosis-related proteins Fas/FasLG and Bax/Bcl-2 while inhibiting the expression of ATF1 and the sperm-associated protein in GC-1 spg cells. DIO and DIO-R could harm sperm production. DIO-R could impair sperm production by inducing the *miR-34c*-activated apoptosis and spermatogenesis pathway, which may be different from that of DIO.

## 1. Introduction

In recent years, the prevalence of infertility has increased gradually, affecting nearly 15% of couples worldwide [1]. The incidence rate of male infertility comprises more than 50% of all infertility. An important reason for infertility in men is related to spermatogenesis in the testes [2]. Recent studies have found that obesity in male mice is related to infertility; the mechanism behind this may be related to a damaged testicular morphology, weakened sperm fertilization ability, etc. [3,4].

There are some people whose weight is normal, but their body fat percentage is as high as that in obese people; this is usually called “masked obesity” [5]. This phenomenon was also found in a mouse model (when fed a high-fat diet, the weight of some mice was higher than that of normal-diet mice, and the weight of some mice was similar to that of normal-diet mice), which has been defined as diet-induced obesity resistance (DIO-R) in many studies [6]. Damaged spermatogenesis has been found in humans with “masked obesity”, in DIO-R mice, and in obese humans and DIO mice in some studies and our prior study [7,8,9,10]. Our prior study found that nearly all of the impairment mechanisms of sperm production in DIO and DIO-R mice were the same. However, a different change in the *miR-34* family among them was found in the epigenetic study in our experiments. Thus, in this study, we attempt to find the different mechanisms between the DIO-R and DIO mice because few studies have focused on it, and DIO-R has been overlooked.

*MiR-34* family applications have been studied and mainly include potential cancer biomarkers and targeted therapies for various diseases [11]. By sequencing the seminal plasma miRNA profile of infertile men, it was found that *miR-34c* decreased significantly in azoospermia; however, it increased in asthenospermia when compared to the control group [12]. The expression of *miR-34c* significantly increased based on semen analysis in moderate oligoasthenoteratozoospermia and non-obstructive azoospermia [13]. Comazzetto et al. found that the deletion of both *miR-34b/c* and *miR-449* loci resulted in oligoasthenoteratozoospermia in mice, and *miR-34bc/449* deficiency impairs both meiosis and the final stages of spermatozoa maturation [14]. Overall, *miR-34c* could play an important role in male reproduction, especially in spermatozoa maturation.

Spermatogenesis plays an important role in sperm maturation, which can be precisely controlled by sophisticated gene expression programs. Studies found that retinoic acid (RA) is necessary to induce meiosis in males, and *RARg* (an RA receptor) is expressed in spermatogonia [15,16,17]. Zhang et al. found that *miR-34c* can regulate mouse embryonic stem cell differentiation into male germ-like cells through *RARg* [17]. *Stra8* (stimulated by retinoic acid gene 8) is an RA-induced gene that plays a vital role in spermatogonial proliferation, differentiation, and meiosis [18], and its deletion results in a complete block of spermatogenesis at the pre-leptotene/zygotene stage [19]. RA autonomously induces meiotic initiation by controlling the RAR-dependent expression of Stra8 [16]. Zfp145, encoding the transcriptional repressor *PLZF*, plays a crucial role in spermatogenesis. *PLZF*’s potential binding sites exist in the promoters of *Stra8* [20]. *PLZF* is a spermatogonia-specific transcription factor in the testes required to regulate the self-renewal and maintenance of the stem cell pool [21]. Overall, we concluded that *miR-34c* may act on *RARg*, further influencing the expression of *Stra8* and, thus, *PLZF* in testicular tissue.

Spermatogenesis is also affected by other factors; Shaha C’s study showed that apoptosis, an ongoing physiological phenomenon, may play a role in male infertility if deregulated, which may lead to the destruction of spermatogenic cells [22]. Apoptosis is also considered the main factor in male reproductive impairment due to diet-induced obesity [3]. Studies have found that the molecular mechanisms involved in the induction of germ cell apoptosis include both intrinsic mitochondrial *ATF1/Bcl2/Bax* and extrinsic *Fas/FasL* pathways [23]. Further, *miR-34c* contributes to cell apoptosis by targeting *ATF1/Bcl2* [24,25,26,27]. Thus, we hypothesize that *miR-34c* may regulate spermatozoa maturation through the *ATF1/Bcl2/Bax* and *Fas/FasL* pathways. Studies have shown that *Fas/Fasl, Bcl-2/Bax, caspase-3*, and *caspase-8* (used to evaluate apoptosis levels) were upregulated in obese groups compared to normal-weight men [28]. Whether this pathway plays a role in DIO and DIO-R mice remains to be verified.

Our prior experiment found differing expressions of *miR-34-5p* between DIO-R and DIO mice. Thus, we want to reveal the possible mechanism of *miR-34-5p* in male reproduction, especially in DIO-R mice, which may differ from DIO mice. In vitro and in vivo experiments were used to verify this. The DIO and DIO-R mouse model was established according to our prior study [29]. *C57BL/6J mice* were fed a high-fat diet for 10 weeks, and the first third of the weight gain was defined as DIO mice, and the last third of the body weight gain was defined as DIO-R mice [6]. GC-1 spg cells were used to disclose the mechanism of sperm production via the induction of *miR-34c*-activated apoptosis and the spermatogenesis pathway.

## 2. Results

### 2.1. Comparison of Body Weight, Adipose Tissue Weight, Visceral Organ Weight, Sperm Count and Motility, and Pathological Change among DIO, DIO-R, and Control Mice

In the 10th week, compared to the control mice, a significant increase in body weight was recorded in DIO mice (*p* < 0.05). Compared to DIO mice, a significant decrease in body weight was recorded in DIO-R mice (*p* < 0.05). No significant difference in body weight was found between the DIO-R and control mice (*p* > 0.05) (Figure 1a). The relative weights of Epi.fat weight in the DIO-R mice were significantly lower than those in the DIO mice (*p* < 0.01); however, no significant difference in Epi.fat weight was found in the DIO-R mice when compared to the control mice (*p* > 0.05) (Figure 1b). The relative weights of Ret.fat weight in the DIO and DIO-R mice were significantly higher than in the control mice (*p* < 0.05) (Figure 1c).

Compared to the control mice, the relative liver weight of the DIO and DIO-R mice decreased significantly (*p* < 0.01) (Table 1). There was no significant difference in the relative kidney and spleen weight among the three groups. We conducted a preliminary assessment of the damage to the reproductive system. As shown in Table 2, the relative weight of the testicular and epididymal tissues significantly decreased in the DIO and DIO-R mice compared with the control mice (*p* < 0.01).

Similarly, the sperm count and motility decreased significantly in both the DIO and DIO-R mice (*p* < 0.01) (Figure 1d,e). We evaluated the pathological changes in the testicular tissue of each group. As shown in Figure 2a, the testicular sections from the control mice with a standard diet displayed normal histology. In contrast, distinct structural disorganization was observed in the DIO and DIO-R mice. In the control group, the testis tissue structure was clear, the spermatogenic cells and supporting cells were arranged regularly, and the seminiferous tubules were normal without obvious lesions. In the DIO group, the number of spermatogenic cells and supporting cells in the testis decreased, the lumen was enlarged (yellow arrow), and some became thinner. The seminiferous tube was deformed, and the cytoplasm was vacuolated (black arrow). The DIO-R mice had similar pathological changes to the DIO mice. Testicular oil red O staining (Figure 2b) showed that the deposition of lipid droplets significantly increased in the DIO mice. In contrast, no obvious deposition of lipid droplets was found in the DIO-R mice.

### 2.2. Comparison of Serum Hormones and Reproduction-Related Enzymes of Testicular Tissues among DIO, DIO-R, and Control Mice

As shown in Table 3, compared to those in the control mice, the serum leptin and E2 levels increased significantly in the DIO and DIO-R mice (*p* < 0.05). However, the level of testosterone and LH significantly decreased in the DIO and DIO-R mice (*p* < 0.05).

HAse, located in the head and acrosome of human sperm, is involved in the fertilization process, giving sperm adhesion and penetration. The activities of acrosome enzyme (ACE) and HAse decreased significantly in the DIO and DIO-R mice compared to the control mice (*p* < 0.01) (Figure 3). Further, the activities of ACE were lower in the DIO-R mice than in the DIO mice (*p* < 0.05).

### 2.3. Comparison of miR-34c and Spermatogenesis-Related Proteins in Testicular Tissues among DIO, DIO-R, and Control Mice

The expression level of *miR-34c* in the testicular tissue was measured using qRT-PCR. Compared with the control group, the expression level of *miR-34c* decreased in the DIO mice, but there was no significant statistical difference. Meanwhile, the *miR-34c* level increased significantly in the DIO-R mice (*p* < 0.01). Further, the expression level of *miR-34c* in the DIO-R mice was significantly higher than that in the DIO mice (*p* < 0.01) (Figure 4a).

As shown in Figure 4b, compared with the control and DIO groups, a significant decrease in retinoic acid receptor γ (RARg), stimulated by retinoic acid gene 8 (Stra8), and promyelocytic leukemia zinc finger (PLZF) expression was observed in the DIO-R mice (*p* < 0.05) (Figure 4c–e). There was no difference in RARg, Stra8, or PLZF expression between the DIO group and the control group (*p* > 0.05).

### 2.4. Comparison of Apoptosis-Related Proteins in Testicular Tissues among DIO, DIO-R, and Control Mice

As shown in Figure 5, compared with the control group, the activating transcription factor 1 (ATF1) expression significantly decreased in the DIO and DIO-R groups (*p* < 0.01). Further, compared with that in the DIO group, the expression of ATF1 significantly decreased in the DIO-R group (*p* < 0.05).

Compared with the control mice, in the DIO mice, the ratio of Bcl-2/Bax significantly decreased, while the cleaved caspase-3 level significantly increased (*p* < 0.05). The B-cell lymphoma-2 (Bcl-2) protein expression level in the testis significantly decreased (*p* < 0.05), but the bcl-2 associated X Protein (Bax) protein expression level increased; thus, the Bcl-2/Bax ratio significantly decreased (*p* < 0.01). However, the factor-related apoptosis (Fas), Fas ligand (FasLG), cleaved caspase-8, caspase-8, cleaved caspase-3, and caspase-3 protein expression levels significantly increased in the DIO-R mice (*p* < 0.05). Compared with the DIO group, the levels of ATF1 and Bcl-2 protein expression and Bcl-2/Bax significantly decreased (*p* < 0.05). However, the protein expressions of Bax, FasLG, and caspase-8 significantly increased (*p* < 0.05) in the DIO-R mice (*p* < 0.05).

### 2.5. Inhibiting Effect of Over-Expression of miR-34c on Sperm-Associated Protein in GC-1 Spg Cells

We detected the key protein expression level of sperm formation via the Western blot method. The levels of RARg, Stra8, and PLZF significantly decreased in the over-expressed *miR-34c* group compared with the control group (*p* < 0.05, Figure 6).

### 2.6. MiR-34c Over-Expression Enhanced Apoptosis in GC-1 Spg Cells

*MiR-34c* mimics were transfected into GC-1 spg cells. A transfection indicator, Negative control FAM (NC FAM), was used to detect the transfection efficiency of GC-1 spg cells. As shown in Figure 7, GC-1 spg cells were treated with *miR-34c* transfection reagent for 24 h. The cells showed a higher green fluorescence, and according to the shooting field of view, the transfection efficiency reached 90% (Figure 7a). qRT-PCR analysis was used to detect the level of *miR-34c*. The *miR-34c* level increased in the over-expressed *miR-34c* group (*p* < 0.01, Figure 7b).

The apoptotic ratio was detected by Annexin V/PI double staining using flow cytometry. Cell populations in the Q2 (upper right) and Q3 (lower right) quadrants of the flow cytometry scatter plot were counted and expressed as percentages of GC-1 spg cells in the late and early apoptosis phases, respectively. Early apoptosis cells included the cell population that was Annexin V-positive and PI-negative (Q3). Late apoptotic cells included a cell population that was both Annexin V- and PI-positive (Q2).

The results showed that the late and early apoptotic ratio in the *miR-34c* over-expression group was significantly higher than in the control group (*p* < 0.01, Figure 7c,d). A confocal laser microscope was used to detect the expression of caspase-3 and caspase-8 expression. The expression of caspase-3 and caspase-8 increased significantly in the *miR-34c* over-expression group (*p* < 0.01, Figure 7e–h).

### 2.7. MiR-34c Over-Expression Inhibited ATF1 Expression and Enhanced Fas/FasLG and Bax/Bcl-2 Expression in GC-1 Spg Cells

As shown in Figure 8, the protein expressions of ATF1 in the *miR-34c* over-expression group were significantly lower than those in the control group (*p* < 0.05) (Figure 8a,b). There was a significant increase in the expression of the key pro-apoptotic protein Bax and a significant decrease in the expression of the key anti-apoptotic protein Bcl-2 in the *miR-34c* over-expression group compared to the control group (*p* < 0.05) (Figure 8a,c–e). Further, the expression levels of Fas and FasLG proteins increased significantly in the *miR-34c* over-expression group compared to the control group (*p* < 0.05) (Figure 8a,f,g); similar results were found in the protein expressions of caspase-8, cleaved caspase-8, caspase-3, and cleaved caspase-3 (Figure 8a,h–k).

## 3. Discussion

In this experiment, the weight of the obese DIO mice increased significantly compared to that of the DIO-R mice. The DIO-R model was successfully established. In the DIO and DIO-R mice, the serum testosterone levels decreased and serum leptin levels increased. Disturbed hormone levels may inhibit sperm counts and increase the proportion of abnormal sperm. Hyaluronidase and acrosomal enzymes are indispensable neutral proteolytic enzymes in the fertilization process. The activity of these two enzymes was reduced in the DIO and DIO-R mice, which may prevent the sperm from completing fertilization properly. These results indicated that a high amount of fat can affect male reproduction, especially spermatogenesis, in DIO and DIO-R mice.

The progressive deterioration of the testicular tissue structure can be shown with HE staining. In this study, pathological injuries of spermatogenic cells and supporting cells in the DIO and DIO-R mice were found. However, the deposition of lipid droplets differed between the DIO and DIO-R mice. We thought that the mechanism of pathological injuries in the testicular tissue of DIO and DIO-R mice may be different.

Male reproductive dysfunction was associated with miRNA problems that strictly regulate these processes [30,31]. Many studies have identified the *miR-34* family as the first miRNAs with important functions in spermatogenesis, and dysregulation was responsible for oligospermia and infertility [32]. In this experiment, qRT-PCR was used to detect the expression of *miR-34c-5p* in the testes of the mice, and the expression level of *miR-34c-5p* in the DIO-R mice significantly increased compared with the control and the DIO mice. Therefore, we believe *miR-34c-5p* plays an important role in male reproductive injury in DIO-R mice. Furthermore, the *miR-34c-5p* downstream regulatory protein was detected in order to explore the possible mechanism of *miR-34c-5p* on male reproductive injury.

RA is synthesized by dedicated enzymes, retinaldehyde dehydrogenases (RALDH), and binds to and activates nuclear *RARa, RARb*, and *RARg* within the RA-synthesizing cells or neighboring cells. RA and its receptors are important factors in spermatogenesis [33] and spermatogonia (at least partially), which are direct targets of RA receptors (including RARg) involved in spermatogonia differentiation [16]. *Stra8* is a tretinoin (RA)-induced gene that plays an important role in spermatogonial proliferation [18]. In addition to its established role as an “amplifier” in meiotic procedures, *Stra8* interacts with different transcription factors and regulates the balance between spermatonic proliferation, differentiation, and the acquisition of meiotic capacity [34]. Zfp145 encodes the transcriptional inhibitor *PLZF*, which plays a key role in spermatogenesis [21]. *miR-34c* can target and downregulate the *RARg* gene, and the potential regulation of *RARg* via *miR-34c* may be related to spermatogenesis [17]. In this study, a decrease in *RARg*, *Stra8*, and *PLZF* expression was observed in DIO-R mice, but not in the DIO group. Thus, in those with DIO-R status, a decrease in *miR-34c* may be due to low levels of *Stra8* and *PLZF*, and impaired spermatogenesis was found. In cell experiments, the over-expression of *miR-34c* could inhibit the expression levels of RARg, Stra8, and PLZF. Thus, we concluded that an increased *miR-34c* level could harm spermatogenesis by influencing spermatogenesis pathway proteins (RARg, Stra8, and PLZF).

Many studies have shown a close relationship between apoptosis and reproductive impairment [3,35]. Many studies have also shown that *miR-34c-5p* is closely related to apoptosis [36,37,38]. This study found that caspase-8, cleaved caspase-3, and cleaved caspase-8 protein levels (reacted apoptosis level) increased in the DIO-R mice, but not the DIO mice. Thus, *miR-34c-5p* may affect reproductive function by affecting the apoptosis pathway. Studies have found that the molecular mechanisms involved in the induction of apoptosis include intrinsic mitochondrial *ATF1/Bcl2/Bax* and extrinsic *Fas/FasL* pathways [23]. We detected the related proteins and found that ATF1 decreased, Bcl-2/Bax increased, and Bcl-2 decreased. Thus, the expression of ATF1 was downregulated, activating the Bcl-2/Bax pathway. The expression of Bcl-2 decreased, Bax expression increased, the Bcl-2/Bax ratio decreased, and the apoptosis of testis tissue increased. Fas and Fas ligands (FasLG) are important pro-apoptotic proteins, and some studies have indicated that the activation of the Fas/FasLG signaling pathway is related to apoptosis in the testes [39]. Variations in FAS function may lead to dysfunctional germline apoptosis mechanisms, resulting in poor ejaculatory sperm quality [17]. In this experiment, the expression of Fas/FasLG in testicular tissues in DIO-R mice was studied after 10 weeks of high-fat-diet feeding, and the expression of Fas/FasLG in the testicles of the DIO-R mice increased. In vitro experiments showed that when *miR-34c-5p* was over-expressed, it inhibited the expression of ATF1, thereby activating the apoptotic pathway Bcl-2/Bax and the Fas/FasLG pathway; inducing the protein levels of the landmark proteins of apoptosis, caspase-3, caspase-8, cleaved caspase-3, and cleaved caspase-8; and promoting increased apoptosis. In summary, we found that the elevated apoptosis levels in DIO-R mice were strongly associated with the high expression of *miR-34c* and the simultaneous activation of these two apoptotic pathways (Fas/FasLG and ATF1/Bcl-2/Bax).

Overall, reproductive damage in the DIO-R mice was closely related to the abnormally high expression of *miR-34c-5p* in testicular tissue. The over-expression of *miR-34c* inhibited the expression levels of RARg, Stra8, and PLZF, which are key proteins in sperm formation, and improved the protein levels of key proteins of apoptosis (caspase-3, caspase-8, cleaved caspase-3, and cleaved caspase-8) by activating the Fas/FasLG and ATF1/Bcl-2/Bax pathways, which induced an increase in apoptosis. This causes spermatogenesis disorders and a reduced sperm count. In summary, this study proves that a high-fat diet may harm the male reproductive system in DIO-R mice. Its mechanism of damage was not completely consistent with that in DIO mice and was closely related to the high expression of *miR-34c-5p* in the testes.

## 4. Materials and Methods

### 4.1. Animal Experiments

#### 4.1.1. Animals and Experimental Design

A total of 40 male *C57BL/6J mice*, 3–4 weeks old, were obtained from the Experimental Animal Center, Shenyang Pharmaceutical University, Shenyang, China. Mice were fed standard laboratory chow for the 1st week to adjust to their new environment. Animals were maintained under standard laboratory conditions on a 12 h light/dark cycle in a temperature-controlled room at 25 ± 2 °C with free access to food and water. All operating procedures in this experiment are in compliance with the guidelines for the care and use of laboratory animals from Shenyang Pharmaceutical University and the guidelines for the care and use of laboratory animals from the National Institutes of Health. All efforts were made to minimize the number of animals used and their suffering (SYPU-IACUC-C2020-8-12-108).

The mice in the control group were fed standard chow (*n* = 10) for 10 weeks. The mice in the high-fat diet group (*n* = 30) received high-fat diet feeding for 10 weeks. The high-fat diet consisted of 60% calories from fat, 19.4% calories from protein, and 20.6% calories from carbohydrates (5.0 kcal/g). The normal diet consisted of 12% calories from fat, 21% calories from protein, and 67% calories from carbohydrates (3.44 kcal/g). The high-fat diet was purchased from TROPHIC Animal Feed High-Tech Co., Ltd. (TP23300, Nantong, China).

After 10 weeks of feeding, according to body weight gain, mice fed the high-fat diet were divided into DIO and DIO-R groups [6]. For the DIO group, the mice exhibiting the first third of the weight gain were defined as diet-induced obesity mice, *n* = 10; for the DIO-R group, mice exhibiting the last third of the body weight gain were defined as diet-induced obesity-resistant mice, *n* = 10. Mice with intermediate tertile body weight gain were excluded from this study. A total of 30 mice (10/group) were included in this experiment.

#### 4.1.2. Sample Collection

The mice were sacrificed using ether anesthesia at the end of the 10th week. Blood samples were collected from the V. cava via a sterile injector and centrifuged at 3000× *g* for 5 min. The serum was separated and then stored at −20 °C until biochemical and hormonal analyses were performed. Testes, epididymides, seminal vesicles, retroperitoneal fat (Ret.fat), and epididymal fat (Epi.fat) were removed and weighed. Testis tissues were fixed in a 4% paraformaldehyde solution for histologic examinations.

#### 4.1.3. Evaluation of Sperm Parameters

The left epididymis was immediately removed after blood sample collection. The epididymis and vas deferens were dissected away from the fat. The epididymis was then cut at the junction between the corpus and cauda epididymis, and the cauda was placed into an Eppendorf tube with 1.0 mL of PBS. Several cuts were made in the cauda epididymis with scissors, and sperm were gently pressed. Then, the sample was allowed to incubate at 37 °C for 4 h to migrate all spermatozoa from epididymal tissue to fluid. Using a hemocytometer (15 mL per side), sperm counts were determined as the number of sperm per microliter, *n* = 10.

Sperm count and motility were assessed in accordance with World Health Organization (WHO) guidelines (≥200 sperm counted for each sample). The sperm count was determined using a hemocytometer. Sperm motility was assessed blindly under a light microscope by classifying 200 sperm per animal as either progressive motile, nonprogressive motile, or immotile. Motility was then expressed as a percent of total motile sperm (progressive motile and nonprogressive motile sperm).

#### 4.1.4. Histopathological Analysis and Oil Red O Staining

Testes were harvested from mice and fixed at 4% paraformaldehyde. Fixed tissues were dehydrated, paraffin-embedded, and sectioned. Sections were deparaffinized and rehydrated before staining. Hematoxylin and eosin (HE) staining was performed for morphological observation using a microscope (Y-TV55, NIKON, Tokyo, Japan), *n* = 3 in each group.

After the testis tissue sections were made, oil red O staining methods were used to observe the changes in lipid droplets. Frozen testis tissue sections were fixed with formaldehyde–calcium, slices were soaked in 60% isopropanol, oil red O staining was performed with a dye solution for 10 min, and differentiation was determined with 60% isopropanol. Oil red O staining was also performed for morphological observation using a microscope (Y-TV55, NIKON, Tokyo, Japan), *n* = 3.

#### 4.1.5. Serum Hormone Analyses

The serum was assayed to determine the leptin and testosterone levels in the mice. Leptin and testosterone (spermatogenesis-related hormones) were detected using an enzyme-linked immunosorbent assay method. The experimental procedure followed the kit instructions. Leptin levels were evaluated with immunoassays (Shanghai Enzyme-linked Biotechnology, Shanghai, China. Cat. No. ml002287), and the lowest detection limit was 0.1 ng/mL. Testosterone levels were assessed with an enzyme-linked immunosorbent assay (Shanghai Enzyme-linked Biotechnology, Shanghai, China. Cat. No. ml001948), *n* = 10.

#### 4.1.6. Testis Tissue Hormone and Enzyme Level Analyses

Ten testicles (*n* = 10) from each group were prepared as a 10% homogenate in order to determine luteinizing hormone (LH), estradiol (E2), hyaluronidase (HAase), and acrosome enzyme (ACE) levels. The LH stimulates testosterone secretion, which plays a role in the promotion of sperm production and maturity. E2, the predominant form of estrogen, plays a critical role in male sexual function. Mammalian fertilization requires sperm to penetrate the cumulus mass and egg zona pellucida prior to fusion with the egg. HAase and ACE have been suggested to participate in penetration events.

LH (Cat. No. ml1063366), E2 (Cat. No. ml001962), HAase (Cat. No. ml037445), and ACE (Cat. No. ml057967) levels were measured using the ELISA method with DRG ELISA kits (Shanghai Enzyme-linked Biotechnology, Shanghai, China), according to the kit manufacturer’s instructions.

### 4.2. Cell Experiments

#### 4.2.1. Cell Culture and Treatment

GC-1 spg cells, a well-established cell line of mouse type B spermatogonia, were obtained from FuHeng BioLogy Company (FuHeng, Shanghai, China, Cat. No. FH1400). The GC-1 spg cells were cultured in DMEM/high-glucose medium (Hyclone, Logan, UT, USA, Cat. No. SH30243.01) supplemented with 10% fetal bovine serum (FBS) (Lonsera, Shanghai, China, Cat. No. LR-S711-001S).

*MiR-34c-5p* mimics (GENERAL BIOL, Xuzhou, China, Order No. RO215228) were used to over-express *miR-34c-5p*. After reaching 60~80% confluence, the cells were exposed to a medium supplemented with GP-transfect-Mate and *miR-34c-5p* mimics. After 24 h, the cells were collected for qRT-PCR or Western blot. The cells were divided into the *miR-34c-5p* over-expression group and the control group.

#### 4.2.2. Apoptosis Assay via Flow Cytometry

GC-1 spg cells were cultured into 6-well plates at a density of 2 × 10^5^ cells/mL, with 4 × 10^5^ cells per well. After transfection, the cells were collected and quantified with an Annexin V-PI apoptosis detection kit. Following the manufacturer’s instructions, the cells were trypsinized and collected via centrifugation at 2000 rpm for 5 min. The cells were resuspended in a 300 μL 1×Binding Buffer supplemented with 5 μL of Annexin V-FITC for 15 min at room temperature in the dark. This was followed by the addition of 5 μL of PI for another 10 min, and then 200 μL of binding buffer was added. Apoptosis was detected using flow cytometry within 1 h, *n* = 3.

#### 4.2.3. Immunofluorescence

GC-1 spg cells were seeded on 24-well culture plates (2 × 10^4^ cells per well), and *miR-34c-5p* mimics were transfected in cells using the above method. The treated GC-1 spg cells were fixed in methanol for 20 min at room temperature, blocked with 3% BSA for 30 min, and permeabilized with 0.25% Triton X-100 in PBS for 10 min. Different protein primary antibodies were incubated overnight at 4 °C. Goat anti-Rabbit IgG/RBITC and DNA dye DAPI were used (blue). A confocal analysis system was used via a laser confocal scanning microscope (C2-si, Nikon, Tokyo, Japan) according to established methods, utilizing continuous laser excitation to minimize the possibility of fluorescence emission leakage, *n* = 3.

### 4.3. Gene Expression Detected by Real-Time Reverse Transcription Polymerase Chain Reaction

miRNA was isolated from the testes or GC-1spg cells using MiPure Cell/Tissue miRNA Kit and DNase treatment. The complementary DNA was synthesized using a miRNA 1st Stand cDNA Synthesis Kit(Nanjing Vazyme Biotech Co., Ltd, Nanjing, China)(by stem-loop) and miRNA-34c-5p primers (U6 mmu-*miR-34c-5p*, reverse-transcription primers: GTCGTATCCAGTGCAGGGTCCGAGGTATTCGCAC-TGGATACGACGCAATC; forward primer: AGGCAGTGTAGTTAGCTGATTG; reverse primer: ATCCAGTGCAGG-GTCCGAGG; U6: forward primer: CTCGCTTCGGCAGCACA; reverse primer: AACGCTTCACGAATTTGCGT), according to the manufacturer’s protocol. For validation, a quantitative real-time polymerase chain reaction was performed on a reverse transcription polymerase chain reaction detection system using miRNA Universal SYBR^®^Green qPCR Master Mix(Nanjing Vazyme Biotech Co., Ltd., Nanjing, China). U6 was used as a housekeeping gene. The determination of gene transcript levels in each sample was performed using the ∆∆ threshold cycle (Ct) method. The Ct of the miRNA was measured and normalized to the average of the housekeeping genes *(*∆Ct = Ct_Unknown_ − Ct_Housekeeping gene_). The fold change of the mRNA in the unknown sample relative to the control group was determined from ∆∆Ct, where ∆∆Ct = ∆Ct_Unknown_ − ∆Ct_Control_. The data are shown as a percentage of the relative miRNA expression to the control group.

### 4.4. Western Blot Analysis

Proteins were extracted from frozen testis tissues and GC-1spg cells using RIPA lysis buffer with PMSF. Protein quantification was measured with a bicinchoninic acid (BCA) protein assay kit. Equal amounts of protein (30 µg) were separated via 10% sodium dodecyl sulfate polyacrylamide gel electrophoresis and were electrotransferred to 0.22 μm polyvinylidene difluoride membranes. After transfer, the membranes were blocked in TBST supplemented with 5% skim milk for 1 h at room temperature and incubated overnight at 4 °C with the relevant primary antibodies (Bax: 1:1000, Cat. No. A19684; Bcl-2: 1:1000, Cat. No. A19693; ATF1:1:1000, Cat. No. A5791; FasLG: 1:1000, Cat. No. A0234; caspase-8: 1:800, Cat. No. A0215; caspase-3: 1:800, Cat. No. A19654; FAS: 1:1000, Cat. No. A0233; RARG: 1:1000, Cat. No. A7448; PLZF: 1:1000, Cat. No. A5863; ABclonal, WuHan, China. Stra8: 1:1000, Cat. No. DF13234, Affinity Biosciences, Changzhou, China). The next day, the membranes were incubated with the HRP-labeled secondary antibodies (1:5000 dilutions) for 1 h at room temperature. Immunoreactive bands were visualized using an ECL reagent under a Gel Imaging System, and protein levels were digitized with ImageJ (1.46R) software. The results of the protein showed only the representative bands. Cleaved caspase-8, caspase-8, cleaved caspase-3, and caspase-3 protein expression levels were detected for apoptosis levels.

### 4.5. Statistical Analysis

All statistical analyses in this experiment were performed with SPSS 24.0 software (SPSS Inc., Chicago, IL, USA). All measurement data are described as the mean ± SD. The analysis of variance (ANOVA) method was used to compare the differences among the three groups, and then the Student–Newman–Keuls (SNK) method was used to compare any two groups. The inspection level was taken at α = 0.05.

## Figures and Tables

**Figure 1 ijms-25-07451-f001:**
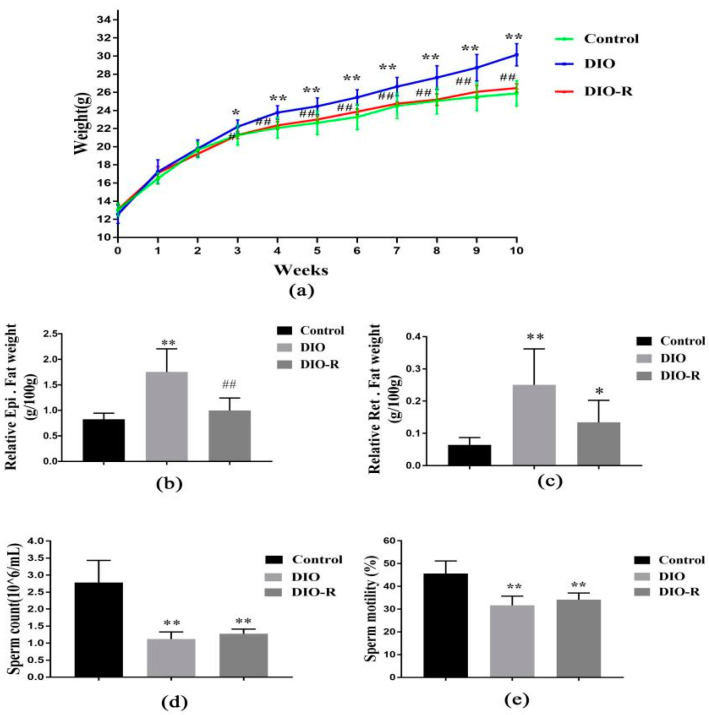
The effect of a high-fat diet on body weight, adipose tissue weight, visceral organ weight, sperm count and motility, and pathological change among DIO-R, DIO, and control mice. (**a**) Body weight; (**b**) relative Epi.fat weight; (**c**) relative Ret.fat weight; (**d**) sperm count; (**e**) sperm motility. Relative Epi.fat weight = epididymal fat weight/body weight × 100; relative Ret.fat weight = retroperitoneal fat weight/body weight × 100; relative Tes. weight = testis weight/body weight × 100; relative epididymis weight = epididymis weight/body weight × 100. All data are expressed as means ± SD, *n* = 10; * *p* < 0.05, ** *p* < 0.01, versus control group; ^#^*p* < 0.05, ^##^ *p* < 0.01, versus DIO group.

**Figure 2 ijms-25-07451-f002:**
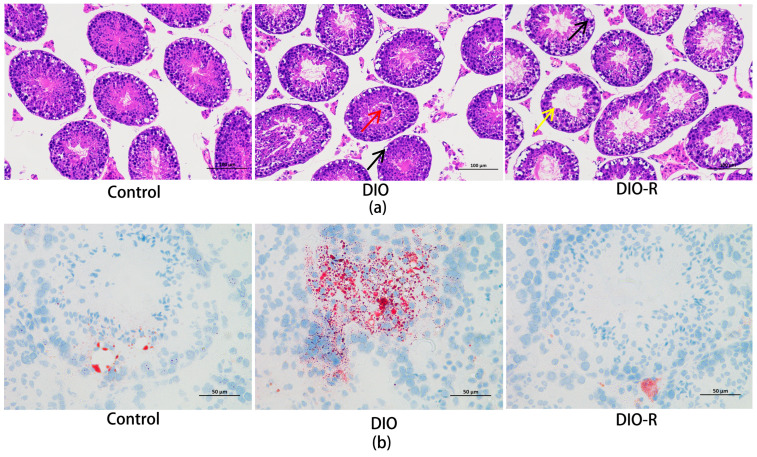
The effect of a high-fat diet on pathological changes among DIO-R, DIO, and control mice. (**a**) Testicular morphology; (**b**) testicular oil red O staining; *n* = 3. A small quantity of spermatozoa shed (red arrow); the lumen was enlarged (yellow arrow); the cytoplasm was vacuolated (black arrow).

**Figure 3 ijms-25-07451-f003:**
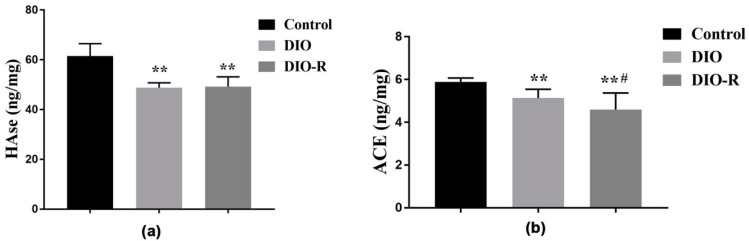
The effect of a high-fat diet on HAse and ACE in testicular tissues among DIO-R, DIO, and control mice. (**a**) HAse level; (**b**) ACE level. All data are expressed as means ± SD, *n* = 10. ** *p* < 0.01 versus control group; ^#^
*p* < 0.05 versus DIO group.

**Figure 4 ijms-25-07451-f004:**
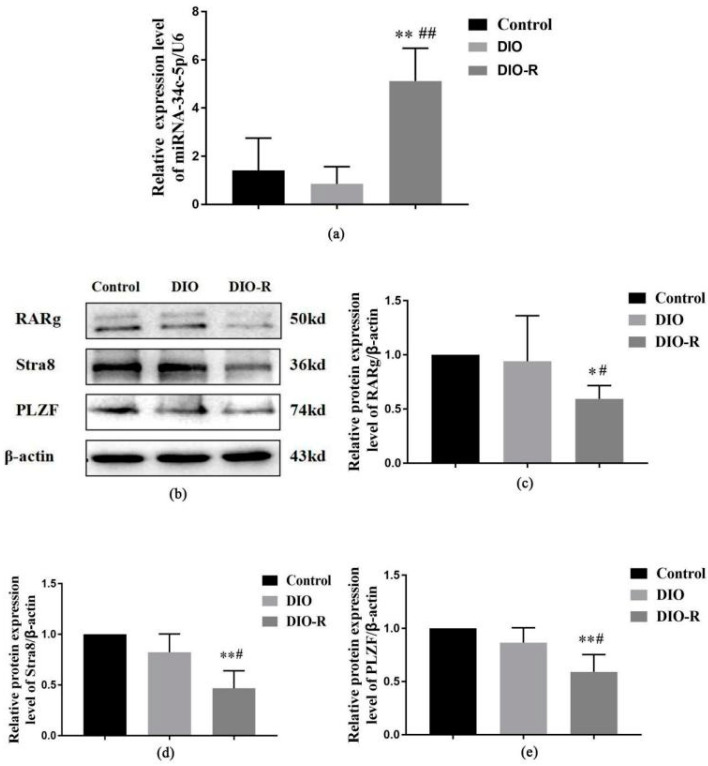
The effect of a high-fat diet on *miR-34c-5p* and spermatogenesis-related proteins in testicular tissues among DIO-R, DIO, and control mice. (**a**) *miR-34c-5p* levels in testicular tissue (*n* = 6); (**b**) representative Western blots using antibodies against RARg, Stra8, and PLZF in testicular tissue (*n* = 6). (**c**–**e**) The bar graphs show the results of the semi-quantitative measurement of RARg, Stra8, and PLZF expression, respectively. All data are expressed as mean ± SD; * *p* < 0.05, ** *p* < 0.01, versus control group; ^#^
*p* < 0.05, ^##^
*p* < 0.01, versus DIO group.

**Figure 5 ijms-25-07451-f005:**
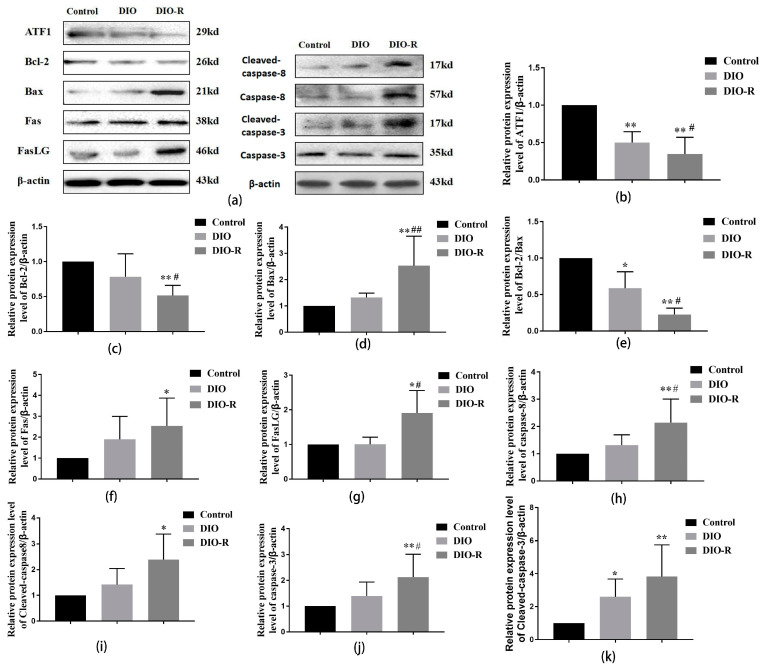
The effect of a high-fat diet on apoptosis-related proteins in testicular tissues among DIO-R, DIO, and control mice. (**a**) Representative Western blots using antibodies against ATF1, Bcl-2, Bax, Fas, FasLG, cleaved caspase-8, caspase-8, cleaved caspase-3, and caspase-3 in testicular tissue; (**b**–**k**) the bar graphs show the results of the semi-quantitative measurement of ATF1, Bcl-2, Bax, Bcl-2/Bax, Fas, FasLG, cleaved caspase-8, caspase-8, cleaved caspase-3, and caspase-3 expression, respectively. All data are expressed as mean ± SD, *n* = 6; * *p* < 0.05, ** *p* < 0.01, versus control group; ^#^ *p* < 0.05, ^##^ *p* < 0.01, versus DIO group.

**Figure 6 ijms-25-07451-f006:**
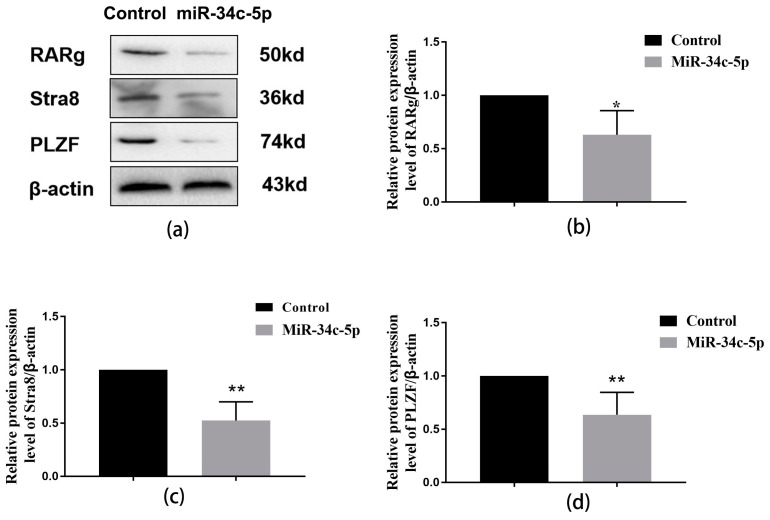
Inhibiting effect of over-expression of *miR-34c* on sperm-associated protein in GC-1 spg cells. (**a**) Representative Western blots using antibodies against RARg, Stra8, and PLZF in GC-1 spg cells; (**b**–**d**) the bar graphs show the results of the semi-quantitative measurement of RARg, Stra8, and PLZF expression, respectively. All data are expressed as mean ± SD, *n* = 6; * *p* < 0.05, ** *p* < 0.01, versus the control group.

**Figure 7 ijms-25-07451-f007:**
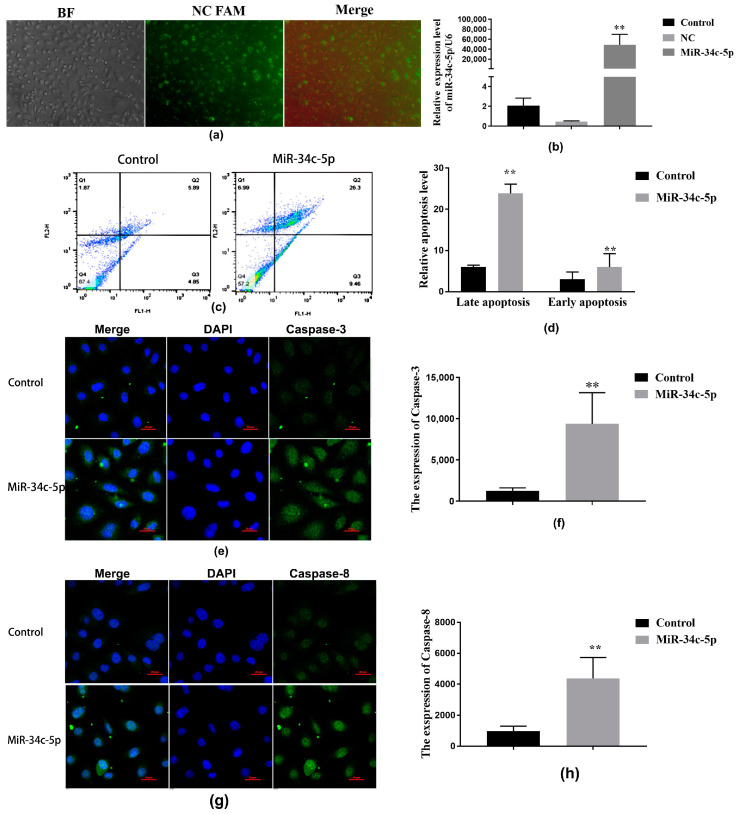
The inhibiting effect of the over-expression of *miR-34c-5p* on sperm-associated protein in GC-1 spg cells. (**a**) A detection chart of the transfection efficiency of GC-1 spg cells (200×), *n* = 3; (**b**) the bar graphs show the level of *miR-34c-5p* in GC-1 spg cells (*n* = 6); (**c**,**d**) the apoptosis level of GC-1 spg cells detected by flow cytometry, *n* = 3. (**e**,**g**) The expression of caspase-3 and caspase-8 detected by a confocal laser microscope, *n* = 6,the scale is 20μm. (**f,h**) The bar graphs show the level of caspase-3 and caspase-8. All data are expressed as mean ± SD; ** *p* < 0.01, versus the control group.

**Figure 8 ijms-25-07451-f008:**
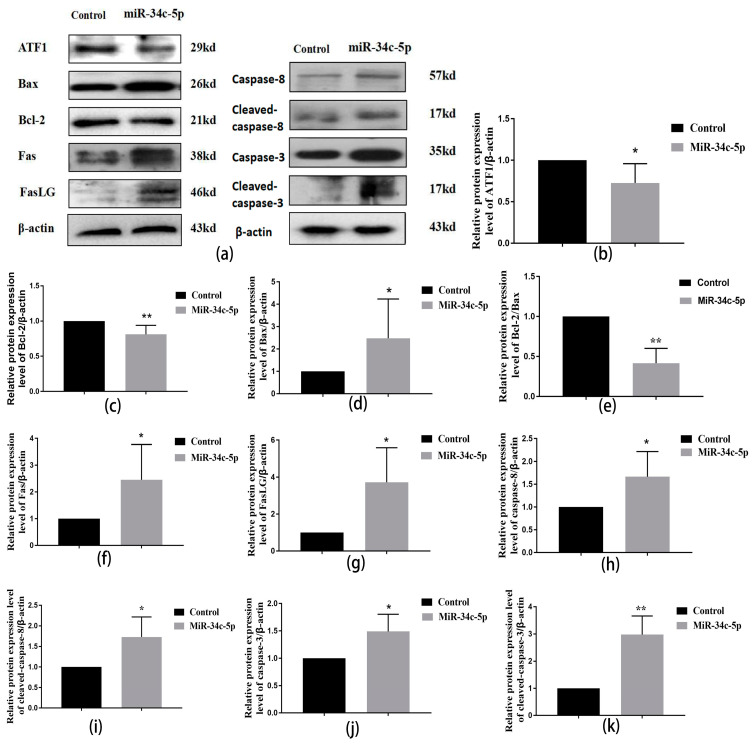
Effect of over-expression of *miR-34c-5p* on ATF1, Bax/Bcl-2, and Fas/FasLG of GC-1 spg cells. (**a**) Representative Western blots using antibodies against ATF1, Bax/Bcl-2, Fas/FasLG, caspase-8, cleaved caspase-8, caspase-3, and cleaved caspase-3 in GC-1 spg cells; (**b**–**k**) the bar graphs show the results of the semi-quantitative measurement of ATF1, Bax/Bcl-2, Fas/FasLG, caspase-8, cleaved caspase-8, caspase-3, and cleaved caspase-3. All data are expressed as mean ± SD, *n* = 6; * *p* < 0.05, ** *p* < 0.01, versus the control group.

**Table 1 ijms-25-07451-t001:** Comparison of the general organs’ relative weight at 10 weeks (mean ± SD, *n* = 10).

Group	Relative Liver Weight (g/100 g)	Relative Kidney Weight (g/100 g)	Relative Spleen Weight (g/100 g)
Control	4.09 ± 0.37	1.10 ± 0.08	0.26 ± 0.04
DIO	3.66 ± 0.28 **	1.17 ± 0.11	0.27 ± 0.06
DIO-R	3.64 ± 0.22 **	1.16 ± 0.09	0.28 ± 0.04

** *p* < 0.01 vs. control group. Relative liver weight = liver weight/body weight × 100. Relative kidney weight = kidney weight/body weight × 100. Relative spleen weight = spleen weight/body weight × 100.

**Table 2 ijms-25-07451-t002:** Comparison of the reproductive organs’ relative weight at 10 weeks (mean ± SD, *n* = 10).

Group	Relative Tes. Weight (g/100 g)	Relative Epididymis Weight (g/100 g)	Relative Sem. Weight (g/100 g)
Control	0.80 ± 0.05	0.30 ± 0.05	0.77 ± 0.11
DIO	0.64 ± 0.10 **	0.25 ± 0.05 **	0.84 ± 0.14
DIO-R	0.69 ± 0.08 **	0.26 ± 0.03 **	0.83 ± 0.12

** *p* < 0.01 vs. control group. Testis—Tes, seminal vesicles—Sem, relative Tes. weight = testis weight/body weight × 100. Relative epididymis weight = epididymis weight/body weight × 100. Relative Sem. weight = seminal vesicles weight/body weight × 100.

**Table 3 ijms-25-07451-t003:** Comparison of hormone levels at week 10 (mean ± SD, *n* = 10).

Group	Leptin (ng/mL)	Testosterone (ng/mL)	E2 (nmol/mg)	LH (mIU/mg)
Control	5.96 ± 0.35	14.74 ± 1.03	14.62 ± 1.00	2.19 ± 0.16
DIO	6.91 ± 0.59 *	13.49 ± 1.38 *	17.47 ± 0.84 **	1.83 ± 0.73 **
DIO-R	6.46 ± 0.64 **	12.51 ± 0.98 **	17.01 ± 6.5 **	1.84 ± 0.16 **

Leptin, testosterone, E2, and LH levels were determined using an ELISA kit (Shanghai Enzyme-linked Biotechnology, Shanghai, China.). * *p* < 0.05 and ** *p* < 0.01 vs. control group.

## Data Availability

The raw data supporting the conclusions of this article will be made available by the authors on request.

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
