# Peer review of "Differential Impairment Mechanism of Sperm Production via Induction of miR-34c-Activated Apoptosis and Spermatogenesis Pathway in Diet-Induced Obesity and Resistant Mice and GC-1 Spg Cells"

_ijms, 2024, doi:10.3390/ijms25137451_

Round 1
Reviewer 1 Report
Comments and Suggestions for Authors
The presented paper is an original paper investigating the impact of high-fat-diet on male fertility using mice model. This issue is interesting and concerns present situation in our society.
The Authors applied many methods but it can make the reader to ask what for? The investigated parameters are not explained enough. There is a story but not clear. The Authors do not connect enough all the results together.
I will start from the end:
It can be concluded that high-fat-diet may impact living organisms differently resulting as clear obesity or mask obesity (obesity resistance) what is somehow in accordance to the literature. The obesity of DIO mice was visible as increase in the body weight and relative increase in retroperotoneal (RET) and epididymal (EPI) fat level in this group of mice however, the total weight of organs decreased (liver, testes and epididymis) or were similar to the Control group (kidney, spleen, seminal vesicles). The gain of fat was also visible in the testes of DIO mice according to the oil red O staining.
The levels of hormones involved in male fertility (leptin, testosterone, LH and E2) were dysregulated in both groups of obese mice but there was no obvious differences between them. When it comes to testicular enzymes ACE and HAse, both enzymes were decreased in both groups of obese mice and the level of HAse was slightly lower in DIO-R than in DIO mice. It may indicate decreased reproductive capacity what was also shown as reduced number and motility of epididymal spermatozoa (especially reduced sperm cell number).
It can also be concluded that the experiments using mice delivered information that DIO-R may have infertility problems bigger than DIO mice as:
- The levels of spermatogenesis-related proteins (AZF-1, RARG, Stra8 and PLZF) were lower in their testes than in the testes from DIO mice;
- The reduced number of testicular cells may be a results of increased apoptosis (indicated as the level of apoptosis-related proteins);
- The visible changes may be a result of increased of miR34c expression;
- The mechanism of miR34c action takes place in germ cells as the dysregulation of fertility-related and apoptotic-related proteins was proven to be present in cell culture of mice spermatogonia (studies using testicles do not precisely indicate the type of cells).
The changes in the testicular histology from mice of both groups are not clear in this paper.
It raises a question if miR34c may drive the mechanisms regulating type of obesity, classical gain of weigh against the obesity resistance? Or maybe rather the type of obesity regulates miR34c expression having worse impact on fertility of obesity resistant males?
The Authors do not present the background of their experimental design enough. Many parameters tested but what for. The Discussion should be a place of the results comparison not explanation of the roles if investigated proteins (at least the main ones). Not the place for results description.
Going into details:
I am not sure if the number presented in Animal experiments is the approval of ethical committee or it is a number of document stating standards in laboratory animal keeping.
It is not known how many samples from each group was included in the experiment. For instance, in histopathology n=3. From each group? Altogether? If from each group, the number of samples is rather low in my opinion. Especially if each group consisted of n=10 mice. It applies to all experiments in this study. It should be clearly indicated in each paragraph in Materials and Methods and under each Figure and Table. This way of presentation of sample number is rather suspicious. If n=3 out of 10, is it a representative result?
2.1.6. It would be easier for the reader if the Author explained shortly in this paragraph (like 1-2 sentences) what all these hormones and enzymes refer to. What was the intention to investigate these parameters?
2.2.2. Flow cytometry - How many cell were seeded in one well? Indicate it in this paragraph.
2.2.3. Immunofluorescnece – which dish was used to prepare microscopic samples? 6-well plate dishes?
2.3. “Experimental methods” – remove this title. All procedures are experimental, flow cytometry too.
When you do not explain what is the connection of these proteins with miR34 in the Introduction, please add at least in 2.3.2 „western blot analysis” paragraph an explanation why you have chosen these proteins. Some of them, like the ones referring to apoptosis, are well-known proteins but the other ones are rather specific. Explain to the reader why. Explain that ATF1 was chosen as a protein connected with MiR-34c, FasLG, FAS, Bax, Bcl-2, Caspase8 and Caspase3 as apoptosis-related proteins and RARG, Stra8 and PLZF as spermatogenesis-related proteins giving a proper references. In the Discussion it can be developed more extensively as it already is however, such information occurring for the first time in the Discussion is too far in the manuscript. If to rearrange the whole paper, the connection of miR34c with AZF1, RARG, Stra8 and PLZF (and even apoptotic proteins) could be explained in the Introduction. Then, it would give background why the Authors decided to investigate it. In the Discussion there should be only information if the observation in the study was similar to other studies and if there were any differences, try to explain why. Maybe not all information should be moved to the Introduction but the most important parts that explain why the Authors investigate them.
Were the samples pooled before loading to the gel? In Figures 7 and 8 there is one band referring to the Control and one referring to miR-34c-5p transfected GC-1 spg cells. In the Figure 7 description, the number of samples is n=6. Does it mean that one band refers to 6 pooled samples or each band refers to 3 pooled samples from one group? In the supporting material there are different combinations of blots following mostly the rule (Control, DIO, DIO-R). Looking at it, it seems that all samples were analysed in separate well. It is not known which Control sample is the reference one. The reference sample should be the same in the all gels. Then it should be stated in the Methods and explained how it was calculated. Add in the Methods that the intensity for each sample was normalized against beta-actin. In addition, the description of Figures presenting blots should indicate that it shows the image of chosen samples. In addition, in my opinion for some cases the image of beta-actin was oversaturated which destroys intensity quantification. The images of blot for Stra8 are of really bad quality…
When it comes to histology of the testicles, in the Results, the Authors wrote that in the both, DIO and DIO-R, groups the pathological changes were similar. Then, in the Discussion, the Authors claim that in DIO mice the injuries were found in supporting cells while, abnormal number and dysfunction (maybe it should be abnormal morphology rather than dysfunction as there was no functional analysis) in DIO-R mice. So how it was? Additionally, in the Discussion the Authors have described their observation what should be in the Results. In the Discussion, the Authors should discuss if similar observations were found in other papers or not. Once again, what means n=3? In my opinion, the n=3 from each group consisting of n=10 mice is too small.
Line 52: “Spermatogenes” or rather “spermatogenesis”?
Line 105: Put the abbreviation for the epididymal (EPI) and retroperitoneal (EPI) fat.
Line 110: The sentence “The epididymal sperm concentration was determined with a hemocytometer” can be removed as it is also stated later in this paragraph.
Paragraph 2.1.4 – the title could be like “ Histology and histochemistry of the testes”. In this paragraph there is no information on oil red O staining procedure but it appears later in the Results.
Figure 7 – the plots h-k are too small. The titles of the plots are not well visible. To be honest, all plots from this Figure could be bigger.
Figure 8 – the images are too small.
Line 382: Please add DIO to make “obese DIO mice” in the sentence.
Paragraph 3.3. Why you have chosen ARF1 to investigate. Explain/ give the reason in the Methods section or even better, connect miR34c with ATF1 in the Introduction. In the literature there is a clear evidence on their connection. Give some background for this.
English requires improvement.
Follow the instructions how to introduce and be consistent with abbreviations.
Summing up, the workflow of this paper is nice but in the presented form it is rather difficult for the reader to understand why this and this factor was investigated. The Authors should work more on the general construction of the paper, although the paragraphs are rather correct. Remember that Introduction give background for your experimental design. In Materials and Methods you can explain your idea adding one sentence like “To check if there is any connection of X with apoptosis we investigated the level of Y and Z”. In Discussion, you do not describe the role of the RNAs/ proteins/ other particles of interest (it should be in Introduction) introducing them to the reader but confront results with the results of others and support Discussion with additional information explaining the observation. The number of samples is sometimes not enough to concern the result as representative for whole group. Altogether, in my opinion, due to the quality of this paper, the Authors should choose different journal for their manuscript.
Comments on the Quality of English LanguageEnglish is understandable although the construction of some sentences is bad some place. It requires moderate improvement.
Author Response
Dear Sir or Madam,
Thank you very much for your comments, positive remarks, and critical points on our manuscript. We have made revisions to the manuscript as you suggested. The changes in the revised manuscript were marked in red. Please allow us to address the issues you raised one by one.

Reviewer 2 Report
Comments and Suggestions for Authors
The authors aimed to explore the role of miR-34c in sperm production in high-fat diet-induced obesity-resistant (DIO-R) mice and GC-1 spg cells, which may differ from high-fat diet-induced obesity (DIO) mice. Clarify the significance of miR-34c.
Could the authors clarify DIO-R animals group. According to the aims of the work, was this previously defined, an obesity resistant group? What is the relevance of this? As observed, both DIO and DIO-R groups impair spermatogenesis. Please explain.
Clearly, miR-34c-5p is associated with apoptosis induction, but the molecular basis of this remains elusive.
Author Response

(The authors gave the same response as above.)

Round 2
Reviewer 1 Report
Comments and Suggestions for Authors
Dear Authors, thank you for your work. The Introduction is rewritten nicely. The new text looks a little bit as a rough package of information and could be more smooth.
The Introduction, line 64 “Study have shown that apoptosis…” What/ whose study? Please, add to the text.
2.1.4, line 116: Write “n=3 in each group”.
The number of animals for testes histology and staining – I do understand the limitations of this procedures and know the difficulties of testes preparation for histology. Still, in my opinion the investigation of 3 samples out of 10 is still not representative and it is not known if the changes are unique to all animals within the same group. One testicle more would be better. In my opinion, this is the weak point of this research but let us leave it as it is.
2.1.5, line 122: Write “Leptin and testosterone (spermatogenesis-related hormones) were detected…”
2.1.6. Write “Ten testicles (n=10) from each group were prepared…” and remove n=10 that is at the end of this paragraph. That is a repetition.
I still do not know which sample is the reference sample in WB experiments. As I wrote, each blot should be run with additional one sample that is repeated throughout all experiments and be treated as “1” value for beta-actin. Then each control sample and other samples should be normalized to it. After that the examined protein should be normalized against relevant beta-actin for each sample. However, maybe I do not understood the way of analysis.
Comments on the Quality of English LanguageStill, English requires improvement.
Author Response
Dear Sir or Madam,
We appreciate the thoughtful review and constructive feedback provided by you.We agree with the your suggestions and will incorporate the recommended changes into the manuscript.The changes in the revised manuscript were marked in red. Please allow us to address the issues you raised one by one.
Reviewers
The Introduction is rewritten nicely. The new text looks a little bit as a rough package of information and could be more smooth.
Thanks for your positive comments for the revision. We revised the introduction. And wish that it looked more smooth. Please find it in Page 2.
The Introduction, line 64 “Study have shown that apoptosis…” What/ whose study? Please, add to the text.
Thanks for your suggestion. I have added the reference for “line 64: Study have shown that apoptosis…”, please find it in Page 2, line 67-69.
2.1.4, line 116: Write “n=3 in each group”.
The number of animals for testes histology and staining – I do understand the limitations of this procedures and know the difficulties of testes preparation for histology. Still, in my opinion the investigation of 3 samples out of 10 is still not representative and it is not known if the changes are unique to all animals within the same group. One testicle more would be better. In my opinion, this is the weak point of this research but let us leave it as it is.
Thank you for your advice, we added “n=3 in each group” in “ 2.1.4. Histopathological analysis and Oil Red O Staining”, please find it in Page 3, line 123-124. As you suggested, one testicle of each mice for histology would be more better. We will increase the number of samples for pathological histology in the next experiments.
2.1.5, line 122: Write “Leptin and testosterone (spermatogenesis-related hormones) were detected…”
Thanks for your suggestion.The “spermatogenesis-related hormones” has been supplemented in Page 3, line 131.
2.1.6. Write “Ten testicles (n=10) from each group were prepared…” and remove n=10 that is at the end of this paragraph. That is a repetition.
Thanks for your remainder. We added “Ten testicles” to make “Ten testicles (n=10)” in the sentence and removed “n=10 (at the end of this paragraph)”, please find it in Page 3, line137.
I still do not know which sample is the reference sample in WB experiments. As I wrote, each blot should be run with additional one sample that is repeated throughout all experiments and be treated as “1” value for beta-actin. Then each control sample and other samples should be normalized to it. After that the examined protein should be normalized against relevant beta-actin for each sample. However, maybe I do not understood the way of analysis.
Thank you for your good advice. As you suggested, each blot should had a reference control “beta-actin”. Usually, the value of protein was analyzed as followed: 1.The expression of target protein in each sample was divided by its internal control “beta-actin”. And the value (the relative expression of target protein in each sample) was obtained after internal parameter correction. 2.Next, the value was used for comparison between samples to obtain the actual change results of the target protein content between different samples, which was a statistical analysis of the ratio difference. Usually other samples could be normalized to control samples (the value of the control samples=1). In general, no additional sample was required. The expression of both the target protein and the internal control protein can be detected in the same time, unless the molecular weights of the target protein and the internal control protein were very close. We wish that we explained the analysis process clearly.
Comments on the Quality of English Language
Still, English requires improvement.
Thank you. The grammar problem was corrected. If required, we'll look for essay polishing from editor to help improve our English writing.